

# Assessing the quality of life among African medical and health science students using the WHOQOL-BREF tool

Fatima Alzahra Galgam[1], Adil Abdalla[2], Mahmoud Shahin[2], Magda Yousif[2], Nahla Abdulrahman[3], Fatmah Alamoudi[2], Mehrunnisha Ahmad[4], Amira Yahia[4], Mohammad Sidiq[5], Aksh Chahal[5], Fuzail Ahmad[6], Mohammad Abu Shaphe[7], Gopal Nambi[8], Moattar Raza Rizvi[9] and Faizan Kashoo[10]

[1] Department of Nursing, International University of Africa, Khourtoum, Sudan
[2] Nursing Department, Prince Sultan Military College of Health Sciences, Dhahran, Saudi Arabia
[3] Nursing College, Najran University, Najran, Saudi Arabia
[4] College of Nursing, Majmaah University, Majmaah, Riyadh, Saudi Arabia
[5] Department of Physiotherapy, School of Allied Health Sciences, Galgotias University, Greater Noida, Uttar Pradesh, India
[6] Respiratory Care Department, College of Applied Sciences, Almaarefa University, Diriya, Riyadh, Saudi Arabia
[7] Department of Physical Therapy, College of Nursing and Health Sciences, Jazan University, Jazan, Saudi Arabia
[8] Department of Health and Rehabilitation Sciences, College of Applied Medical Sciences, Prince Sattam Bin Abdulaziz University, Al-kharj, Saudi Arabia
[9] College of Healthcare Professions, Dehradun Institute of Technology (D.I.T) University, Makka Wala, Uttarakhand, Dehradun, India
[10] Department of Physical Therapy and Health Rehabilitation, College of Applied Medical Sciences, Majmaah, Riyadh, Saudi Arabia

Corresponding author
Faizan Kashoo, f.kashoo@mu.edu.sa

## ABSTRACT

**Background:** The quality of life (QoL) among health professional students is available in the literature, yet there is a paucity of information concerning QoL among African students. The study aimed to measure the QoL with the World Health Organization Quality of Life-BREF (WHOQOL-BREF) tool among African medical and health science students.

**Methods:** A cross-sectional study was conducted involving 349 African medical and health science students from various disciplines at the International African University in May 2024. A purposive sampling method was used to recruit participants from five different faculties. Data were collected using the WHOQOL-BREF tool, and analyzed using descriptive statistics, chi-square test and multiple linear regression to determine the predictors of QoL among students.

**Results:** The overall QoL among 349 African medical and health science students was moderate, with a mean score of 67.5% ± 10.8%. The highest mean scores were observed in the physical health domain (69.3% ± 12.0%), while the lowest scores were in the environmental domain (62.9% ± 12.0%). The multiple regression analysis using demographic data as predictors of QoL revealed that dentistry students were significant predictors of higher overall QoL scores compared to other student groups (β = 7.059, $p < 0.05$), as well as specific QoL domains including physical health (β = 6.328), psychological health (β = 8.415), social relationships (β = 7.823), and environment (β = 7.017). Furthermore, students from the fields of laboratory sciences

and medicine significantly predicted higher scores in the physical health domain
($\beta$ = 5.223) and the psychological health domain ($\beta$ = 4.433), respectively. Age was
also a significant predictor; students aged between 20 and 23 years showed a positive
impact on social relationship domain of QoL ($\beta$ = 10.296). However, second year
($\beta$ = −11.146), third year ($\beta$ = −13.629), and fourth-year students ($\beta$ = −10.144)
exhibited lower social relationship domain of QoL scores.

**Conclusion:** Students of medical and health sciences in Africa exhibited moderate
quality of life (QoL). The findings indicate that dentistry students generally
experience higher QoL across multiple domains, which contrasts with students from
other disciplines, such as pharmacy and nursing. Age and academic year were also
significant predictors of QoL, with younger students and those in their initial years of
study reporting lower scores. These results align with existing literature and
underscore the need for targeted interventions to support students, particularly those
in high-stress disciplines or at earlier stages of their education.

**Subjects** Psychiatry and Psychology, Science and Medical Education, Mental Health, Healthcare
Services
**Keywords** Quality of life, Medical students, Cross-sectional study, Africa

## BACKGROUND

Quality of life (QoL) is an important multidimensional construct consisting of four main
domains (physical, psychological, emotional, social and environmental domains) (*Miguel
et al., 2021*). Defined by the World Health Organization (WHO) as individuals' perception
of their life circumstances within cultural and value frameworks (*Costa et al., 2021*). The
QoL determines the overall well-being and productivity of an individual within a society
(*Maridal, 2017*). In medical education, students face challenges and stressors affecting their
QoL (*Steiner-Hofbauer & Holzinger, 2020*). In Africa, lack of infrastructure (*Abdalla,
Omar & Badr, 2016*), shortage of supplies (*Oleribe et al., 2019*), high disease burden
(*Gouda et al., 2019*) makes medical education more challenging (*Auf et al., 2018*). While
research extensively investigates the impact of medical education among western students'
mental health and well-being (*Ribeiro et al., 2018*; *Feng et al., 2019*; *Byrnes et al., 2020*),
there remains a noticeable gap in understanding the current state of QoL among African
students pursuing medical and health science education.

Utilizing the World Health Organization Quality of Life-BREF (WHOQOL-BREF)
survey (*The WHOQOL Group, 1998a*), a validated instrument (*Malibary et al., 2019*) that
offers a multidimensional perspective on life experiences and well-being (*Seok et al., 2023*).
Previous studies have utilized the WHOQOL-BREF to evaluate quality of life among
diverse populations, including students in healthcare disciplines (*Alkatheri et al., 2020*).
For instance, a study conducted among medical students in Saudi Arabia found significant
associations between academic stressors and lower QoL scores (*Mahmoud & Fareed,
2018*). Similarly, research in China using the WHOQOL-BREF demonstrated variations in
QoL across medical, preventive medicine, and nursing students (*Li et al., 2020*).
Additionally, the WHOQOL-BREF's robust psychometric properties, including high
internal consistency across multiple cultures, make it suitable for the African student

population (*Seok et al., 2023*; *Ibrahim et al., 2024*). The WHOQOL-BREF was specifically chosen for this study due to its multidimensional approach to assessing quality of life, which encompasses physical health, psychological health, social relationships, and environmental factors. It is a validated tool that has been used extensively across various populations, making it highly applicable for evaluating quality of life among diverse groups of *The WHOQOL Group (1998a)*. The WHOQOL-BREF also provides the advantage of cultural adaptability, which is particularly important given the diverse backgrounds of the students at the International University of Africa.

Other tools, such as the SF-36 (*Lin et al., 2020*) and the EQ-5D (*Feng et al., 2021*), were considered. The SF-36 is commonly used for health-related quality of life assessment, but it is primarily focused on physical and mental health components and lacks specific environmental and social domains that are crucial for understanding the broader quality of life of students (*Busija et al., 2020*). The EQ-5D, while being simpler and faster to administer, is more appropriate for assessing health status rather than a holistic quality of life. Given the scope of this study, which aims to understand multiple facets of quality of life in the context of a challenging academic environment, the WHOQOL-BREF was considered the most suitable tool.

The rationale for this study lies in the growing recognition of the importance of QoL as an indicator of well-being among health professional students, who often face unique stressors related to their demanding academic environment (*Busija et al., 2020*). In Africa, these challenges are compounded by systemic issues such as limited resources, high disease burden, and socio-economic constraints, which can significantly impact students' QoL (*Bashir et al., 2020*; *Busija et al., 2020*). While similar studies have been conducted in Western countries, there is a noticeable gap in the literature regarding QoL among African health science students. This study is important because it aims to fill this gap by providing data on the QoL of African medical students, using a validated and culturally adaptable tool (WHOQOL-BREF). The insights gained can help inform targeted interventions to support student well-being, ultimately contributing to better academic performance and the future health workforce in Africa. Based on the identified gaps in the literature and the need to better understand the quality of life among African health science students. This study aims to answer the following research question: What are the key demographic and academic factors that influence the QoL among African medical and health science students as assessed by the WHOQOL-BREF tool? We aim to provide insights into the factors shaping the QoL of students in Africa. We hypothesize that African students enrolled in medical and health science programs experience variations in their QoL across different domains, influenced by various demographic variables. We hope to inform the development of targeted support services and curriculum enhancements that foster holistic student development.

# MATERIALS AND METHODS

## Study design

This is a descriptive cross-sectional institutional-based study aimed at assessing the quality of life among students in the Medical and Health Sciences faculties at the International

University of Africa (IUA). Ethical approval was obtained from the Research Ethical Committee (REC) of the Faculty of Medicine at IUA (IRB-21).

### Study area/setting

The study was conducted at the IUA, located in Khartoum State, Sudan. IUA has a diverse student body of approximately 10,400 students, representing 84 different nationalities. The university includes various faculties and programs, including applied and humanitarian faculties. This study focused on the medical and health science faculties: Medicine, Nursing, Pharmacy, Medical Laboratory, and Dentistry with 3,000 students.

### Inclusion and exclusion criteria

The study included students enrolled in the faculties of Medicine, Nursing, Pharmacy, Medical Laboratory, and Dentistry at the International University of Africa, who were aged 18 years or older and provided informed consent to participate. Students who were not enrolled in these specific faculties, were under the age of 18, or who did not provide consent were excluded from the study. Additionally, students who were unable to complete the questionnaire due to language barriers or health issues were also excluded to ensure data quality and reliability.

### Study population

The study was conducted from May 1st to May 19th, 2024. This cross-sectional study was conducted at the IAU in Khartoum, Sudan, specifically within the faculties of Medicine, Dentistry, Nursing, Pharmacy, and Medical Laboratory Sciences were the medium of instruction is English. We choose cross-sectional studies because they are widely used to assess the QoL in student populations, as they provide a snapshot of health or well-being at a single point in time. Several studies, such as those by *Pagnin & De Queiroz (2015)* and *Li et al. (2020)*, have successfully employed cross-sectional designs to explore QoL and its associations with academic factors. Thus, cross-sectional study design was quick, cost-effective, allow large sample sizes, help assess the prevalence of QoL issues, compare subgroups within African students, generate hypotheses for future research, and involve minimal attrition. The curriculum durations were as follows: Medicine and Dentistry programs each extend over 5 years with a two-semester system per year, while the Nursing, Pharmacy, and Medical Laboratory Sciences programs are each structured as four-year courses. The program follows clinical posting of students from first year of training except for pharmacy faculty. The objectives of the study were clearly explained to the participants through the questionnaire, ensuring the confidentiality of their information. The original English version of WHOQOL-BREF was sent to the participants with through email and other electronic medium. The questionnaire began with an electronic consent form, reassuring participants of their autonomy to withdraw from the survey at any time.

### Sample size and sampling technique

The sample size calculation was performed using Epi Info software, a tool developed by the Centers for Disease Control and Prevention (CDC) (*Dean et al., 2000*). With a population size of 3,000 African medical and health science students and a desired confidence level of

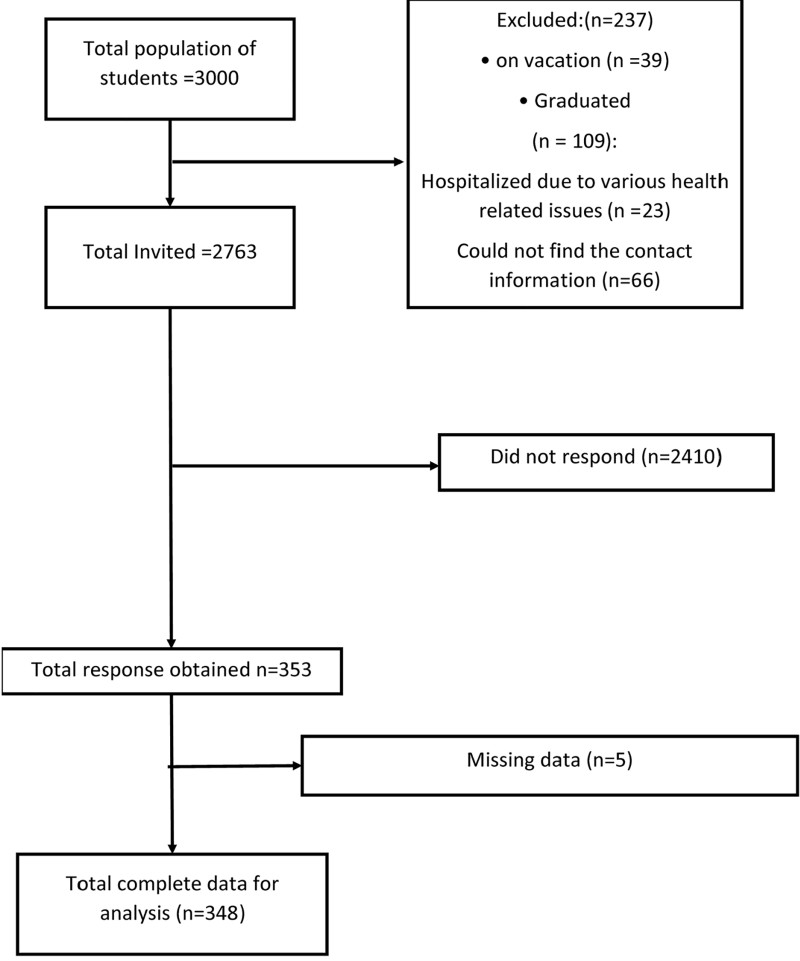

**Figure 1 Flow chart of total population, invitation and reason of drop out of medical and health science students.**

95% with a 5% margin of error, the software determined that a sample size of approximately 350 participants would be sufficient to ensure statistical validity and reliability in assessing the quality of life among these students. A non-probability sampling technique, specifically purposive sampling, was employed to recruit students for the medicine and other health science faculty (Fig. 1). This method was chosen because it allowed us to focus on specific subgroups within the health sciences that were of interest for assessing quality of life. By using purposive sampling, we were able to ensure that participants from each relevant discipline were adequately represented, thereby providing insights into the differences in quality of life across these disciplines. The contact details of the participants were obtained from the registration section of the IUA.

## Data collection tool
The WHOQOL-BREF tool, developed by the World Health Organization (WHO), was used for data collection, and formal permission was obtained from

WHO to use the tool (*The WHOQOL Group, 1998b*). The questionnaire consisted of two parts:

1) Socio-demographic data: This section gathered basic information about the participants.
2) Quality of life assessment: This section included 26 items across four domains—physical health, psychological health, social relationships, and environment. Each item was rated on a 5-point Likert scale ranging from 1 (very poor/very dissatisfied/never/none) to 5 (very good/very satisfied/always/extremely). Additionally, two items assessed the overall satisfaction with the students' quality of life. The scoring was converted to 0–100 scoring through a standard guideline (*World Health Organization, 1996*). The scoring was further categorized as excellent quality of life (70–100%), moderate quality of life (50–69%), poor quality of life (30–49%), very poor quality of life (0–29%).

Reliability: The internal consistency of the 26-item WHO-QoL scale, as evaluated by Cronbach's alpha coefficient, was 0.904. The coefficient for the Physical Health domain (Q3, Q4, Q10, Q15, Q16, Q17, and Q18) was 0.715, showing a slight improvement to 0.735 upon the deletion of Q3, which pertains to physical pain. In the psychological domain (Q5, Q6, Q7, Q11, Q19, and Q26), the coefficient was 0.713, showing a slight increase to 0.746 after removing Q26, which was associated with negative feelings. The social relationship domain (Q20, Q21, and Q22) had a coefficient of 0.663, with no observed improvement upon removing any item. Similarly, the Environment domain (Q8, Q9, Q12, Q13, Q14, Q23, Q24, and Q25) showed a coefficient of 0.773, with no improvement noted upon removing any item.

## Data collection technique

Data were collected through a self-administered, close-ended questionnaire, formatted in Google Forms and distributed *via* various online platforms, including WhatsApp and Telegram.

## Statistical analysis

The statistical analysis encompassed three primary components. Firstly, the socio-demographic characteristics of the sample were summarized using frequency, percentage, mean, and standard deviation. Demographic data were analyzed using a one-sample binomial test for dichotomous variables and a one-sample chi-square test for categorical variables with more than two levels. Internal consistency was then assessed through the computation of Cronbach's alpha coefficient. Addressing missing data involved replacing them with the median value derived from all respondents for the respective question. Any missing value for a domain beyond 20% was subsequently removed to maintain data integrity. To compare the QoL among college students across different groups based on variables such as years, specialties, and gender, multiple linear regression analysis was employed. All key assumptions for linear regression were tested and satisfied. Linearity was confirmed through scatter plots and partial regression plots. The independence of residuals was validated using the Durbin–Watson test.

**Table 1  Demographic characteristics of participants.**

| Variable | Category | Frequency | Percent | WHO-QoL BREF mean (SD) | p-value |
|---|---|---|---|---|---|
| Gender | Male | 135 | 38.7 | 68.1 (11.5) | 0.001* |
| | Female | 214 | 61.3 | 67.1 (10.3) | |
| Nationality | East Africa | 171 | 49 | 68.0 (11.3) | 0.001** |
| | West Africa | 73 | 20.9 | 67.6 (10.6) | |
| | North Africa | 93 | 26.6 | 66.5 (9.8) | |
| | Others | 12 | 3.4 | 68.0 (11.4) | |
| Age group | <20 years | 40 | 11.5 | 63.9 (10.9) | 0.001** |
| | 20–23 years | 151 | 43.3 | 67.3 (10.7) | |
| | >23 years | 158 | 45.3 | 68.6 (10.7) | |
| Marital status | Single | 294 | 84.2 | 67.4 (10.9) | 0.001** |
| | Married | 45 | 12.9 | 69.2 (9.5) | |
| | Divorced | 10 | 2.9 | 62.6 (10.1) | |
| Faculty | Pharmacy | 60 | 17.2 | 64.9 (11.1) | 0.001** |
| | Nursing | 94 | 26.9 | 67.0 (9.2) | |
| | Medicine | 119 | 34.1 | 67.7 (9.2) | |
| | Laboratory | 37 | 10.6 | 68.6 (12.0) | |
| | Dentistry | 39 | 11.2 | 71.7 (9.4) | |
| Year/Level | First year | 38 | 10.9 | 65.8 (10.7) | 0.001** |
| | Second year | 39 | 11.2 | 65.1 (11.4) | |
| | Third year | 67 | 19.2 | 65.8 (10.7) | |
| | Fourth year | 138 | 39.5 | 68.4 (10.6) | |
| | Fifth year | 67 | 19.2 | 69.7 (10.5) | |

Note:
WHO-QoL BREF, World Health Organization Quality of Life Scale Brief Version; $p^*$, One sample binomial test; $p^{**}$, Chi-square test.

Homoscedasticity was assessed with the Breusch–Pagan test and residual plots. The normality of residuals was verified using Q-Q plots, the Shapiro–Wilk test, and histograms. Multicollinearity was checked with variance inflation factors (VIFs) and tolerance levels. Finally, outliers and influential points were identified and managed through Cook's distance and leverage values. All statistical analyses were carried out using SPSS software (version 20.0; SPSS Inc., Chicago, IL, USA) for Windows. A significance level of $p < 0.05$ was employed to determine statistical significance.

## RESULTS

### The sample's demographic characteristics

The study sample consisted of 349 African medical and science college students from the International University of Africa, representing 11.6% of the total student population ($n = 3,000$). The sample comprised $n = 135$ (38.7%) males and $n = 214$ (61.3%) females. The majority of student's hail from East Africa ($n = 171$, 49.0%), while only 12 (3.4%) are from other regions (Chad, Ethiopia, Egypt, India, and Mozambique). Regarding age distribution, most students fell within the 20–23 years' category ($n = 151$, 43.3%), followed

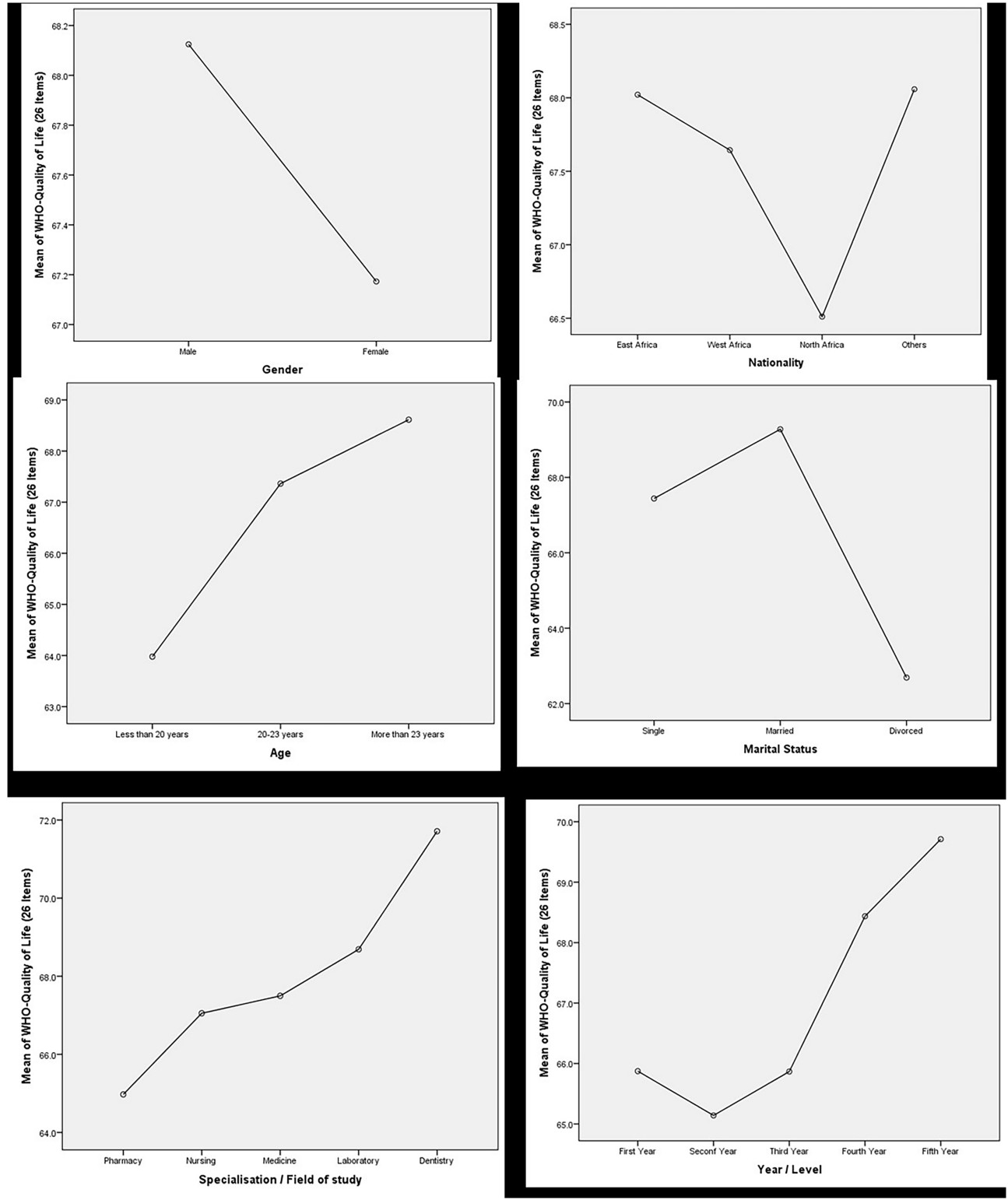

**Figure 2  Relationship between demographic variables and mean score of WHOQOL-BREF tool.**

**Table 2 Multiple linear analysis.**

| Dependent variable | Independent variable | $R^2$ | Adj-$R^2$ | RMSE | $F$-value | $B$ | SE | 95% CI | | $T$ Value | $p$-value |
|---|---|---|---|---|---|---|---|---|---|---|---|
| | | | | | | | | Lower | Upper | | |
| Overall QoL | | 0.068 | 0.023 | 10.694 | 1.51 | | | | | | $p = 0.094$ |
| | Dentistry | | | | | 7.059 | 2.295 | 2.545 | 11.573 | 3.076 | $p = 0.002$ |
| Physical health domain of QoL | | 0.046 | 0.001 | 12.046 | 0.996 | | | | | | $p = 0.040$ |
| | Laboratory | | | | | 5.223 | 2.59 | 0.128 | 10.318 | 2.016 | $p = 0.045$ |
| | Dentistry | | | | | 6.328 | 2.585 | 1.243 | 11.413 | 2.448 | $p = 0.015$ |
| Psychological domain of QoL | | 0.075 | 0.031 | 13.198 | 1.685 | | | | | | |
| | Medicine | | | | | 4.433 | 2.196 | 0.112 | 8.753 | 2.018 | $p = 0.044$ |
| | Dentistry | | | | | 8.415 | 2.832 | 2.844 | 13.987 | 2.971 | $p = 0.003$ |
| Social relationship domain of QoL | | 0.072 | 0.027 | 16.043 | 1.601 | | | | | | |
| | Age Group (20–23 year old) | | | | | 10.296 | 3.92 | 2.584 | 18.008 | 2.626 | 0.009 |
| | Age Group (more than 23 years old) | | | | | 8.845 | 4.048 | 0.03 | 0.882 | 2.185 | 0.030 |
| | Dentistry | | | | | 7.823 | 3.443 | 1.051 | 14.595 | 2.272 | 0.024 |
| | Second year student | | | | | −11.146 | 4.084 | −19.18 | −3.112 | −2.729 | 0.007 |
| | Third year students | | | | | −13.629 | 4.277 | −22.043 | −5.215 | −3.186 | 0.002 |
| | Fourth year students | | | | | −10.144 | 3.993 | −17.998 | −2.289 | −2.54 | 0.012 |
| Environment domain of QoL | | 0.066 | 0.021 | 11.971 | 1.461 | | | | | | |
| | Dentistry | | | | | 7.017 | 2.569 | 1.964 | 12.07 | 2.732 | 0.007 |

**Note:**
Adj-$R^2$, adjusted R-square; RMSE, root mean square error; SE, standard error; CI, confidence interval; B, standardized coefficients; QoL, quality of life.

by those aged over 23 years ($n = 158$, 45.3%), with the fewest below 20 years ($n = 40$, 11.5%). In terms of marital status, the majority were unmarried ($n = 294$; 84.2%), while the minority were divorced ($n = 10$; 2.9%). Furthermore, a predominant number of students belonged to the medicine department ($n = 119$, 34.1%), with the dentistry department comprising the smallest proportion ($n = 39$, 11.2%). In terms of academic year, the majority of students were in their fourth year, while the first-year cohort represented the smallest group ($n = 38$, 10.9%) (Table 1).

Differences in QoL with field of study, academic year, gender, marital status, and region. The overall QoL among African medical students showed moderate levels ($M = 67.5\%$, $SD = 10.8$), with the physical health domain scoring the highest ($M = 69.3\%$, $SD = 12.0$) and the environmental domain the lowest ($M = 62.9\%$, $SD = 12.0$). The variation of QoL with demographic data is visually depicted in Fig. 2.

The multiple linear regression model explained 6.8% of the variance in overall QoL scores ($R^2 = 0.068$), though it was not statistically significant ($F (16, 332) = 1.51$, $p = 0.094$). Dentistry students were notably significant predictors of higher QoL ($\beta = 7.059$, $p = 0.002$). In the physical health domain, the model accounted for 4.6% of the variance ($R^2 = 0.046$), but was not significant ($F (16, 332) = 0.996$, $p = 0.461$); however, both laboratory ($\beta = 5.223$, $p = 0.045$) and dentistry students ($\beta = 6.328$, $p = 0.015$) scored significantly higher. The psychological health domain was more robust, explaining 7.5% of the variance ($R^2 = 0.075$) and reaching statistical significance ($F (16, 332) = 1.685$, $p = 0.048$), with

dentistry ($\beta$ = 8.415, $p$ = 0.003) and medicine students ($\beta$ = 4.433, $p$ = 0.044) scoring higher. The social relationships domain accounted for 7.2% of the variance ($R^2$ = 0.072), marginally significant ($F$ (16, 332) = 1.601, $p$ = 0.067), where dentistry students ($\beta$ = 7.823, $p$ = 0.024) and those aged 20–23 ($\beta$ = 10.296, $p$ = 0.009) showed higher scores, in contrast to lower scores from second ($\beta$ = −11.146, $p$ = 0.007), third ($\beta$ = −13.629, $p$ = 0.002), and fourth-year students ($\beta$ = −10.144, $p$ = 0.012). The environmental domain had a variance of 6.6% ($R^2$ = 0.066) without significant change ($F$ (16, 332) = 1.461, $p$ = 0.112), yet dentistry students still scored higher ($\beta$ = 7.017, $p$ = 0.007) (Table 2).

## DISCUSSION

The QoL of 349 African medical students, predominantly female and from the East African region, was evaluated using the WHOQOL-BREF tool, a 26-item questionnaire covering four domains. Most students achieved an overall mean score between 62.9% and 69.3%. The internal consistency of the WHOQOL-BREF tool was excellent. Key demographic factors associated with better QoL included older age group, studying dentistry.

The overall QoL among African students was moderate, characterized by relatively lower scores in the psychological and environmental domains. These findings are consistent with a prior study comparing medical students to a normal age-matched group, which highlighted the psychological and social relationship domains as being the most affected (*Pagnin & De Queiroz, 2015*). Students aged between 20 and 23 years reported higher QoL scores compared to those under 20 years, particularly in the psychological and environmental domains. This suggests that older students may possess more developed coping mechanisms, resilience, or life experience that buffers against the stresses of medical education. However, a study conducted among 116 women at the University of Physical Education in Warsaw revealed a negative correlation between the psychological domain and advancing age (*Niedzielska et al., 2017*). This suggests that gender and field of study might influence the QoL differently among students.

Married students reported better psychological well-being than divorced students, emphasizing the role of social support and stability in enhancing QoL. These findings are supported by a research article conducted among 983 medical students in Saudi Arabia reported that students who lived with their family had better QoL then who lived in hostel or alone (*Mahmoud & Fareed, 2018*).

Field of study comparisons revealed that dentistry students reported significantly higher QoL compared to students in departments. This discrepancy might be due to the unique stressors associated with medicine, nursing, laboratory and pharmacy education, such as higher workload and patient interactions. On the contrary a study conducted in Vietnam among 201 dental students revealed lower QoL as compare to dental students from US and Pakistan (*Vo, Tran & Dinh, 2020*).

Gender differences in QoL were not as pronounced as expected, suggesting that both male and female students experience similar levels of stress and challenges. However, a systematic review of Brazilian medical students identified a negative impact of female gender on the quality of life (*Solis & Lotufo-Neto, 2019*). The lack of significant differences

across academic years, except for a higher representation in the fourth year, suggests that QoL might not vary drastically with academic progression. This could indicate that the stress and challenges of medical education are relatively constant throughout the years of study, or that students develop coping mechanisms that stabilize their QoL over time.

In comparing our findings with previous studies, we observed several similarities and differences across the different QoL domains assessed. Our results showed that dentistry students scored significantly higher in all QoL domains compared to other health science students, which aligns with findings from a study conducted among dental students in Saudi Arabia, where dentistry students reported better physical and psychological health (*Mahmoud & Fareed, 2018*). However, contrasting results were found in a study conducted in Vietnam, where dental students had lower QoL scores compared to medical students, emphasizing the unique context and stressors that can influence student well-being differently (*Vo, Tran & Dinh, 2020*).

In the physical health domain, our findings revealed that students from the laboratory sciences also had significantly higher scores, similar to what was reported among Brazilian medical students reported that physical health domain significantly affected the QoL (*Miguel et al., 2021*). The psychological health domain in our study demonstrated a significant association with the field of study, with dentistry and medicine students reporting better scores. A study conducted in west indies among dental and medical students reported better academic performance with higher subjective wellbeing (*Chattu et al., 2020*).

The social relationships domain in our study showed that students aged between 20 and 23 years reported higher QoL scores, a pattern consistent with prior research from Poland, where younger students were found to have higher scores in social relationships, likely due to greater peer support networks (*Niedzielska et al., 2017*). Interestingly, our study found no significant differences in QoL based on gender, which contrasts with findings from Brazil, where female medical students reported lower QoL scores due to higher levels of stress and anxiety (*Solis & Lotufo-Neto, 2019*).

In the environmental domain, our findings indicated that dentistry students had higher scores compared to students in other faculties, which aligns with the results from previous studies showing that dental programs often provide more structured environments, contributing positively to students' perceived QoL (*Andre, Pierre & McAndrew, 2017*). However, it is important to note that the environmental domain often reflects external factors beyond academic settings, suggesting that students from different regions or institutions may experience variations in QoL due to differing infrastructural support.

## Implications for interventions

The study highlights several areas where targeted interventions could improve QoL among African medical students. Developing support systems for fresher students and those in high-stress specialties like pharmacy and nursing could mitigate some of the negative impacts on their QoL. Programs that enhance coping skills, resilience, and provide psychological support could be particularly beneficial.

Additionally, the high mean score but not significant role of marital status in psychological well-being suggests that fostering strong social support networks within the university might help unmarried students enhance their QoL. Peer support groups, mentorship programs, and counseling services addressing specific needs and challenges faced by medical students could be instrumental.

### Limitations and future research

This study has several limitations. The cross-sectional design provides a snapshot of QoL at a single point in time, limiting the ability to infer causal relationships. Longitudinal studies would be beneficial to track changes in QoL over time and identify key influencing factors. Reliance on self-reported data introduces the potential for response bias. Incorporating objective measures of well-being, such as academic performance or physiological indicators of stress, could provide a more comprehensive assessment of QoL. Future research should explore the impact of specific stressors unique to the African context, such as economic constraints, political instability, and cultural factors, on medical students' QoL. Comparative studies across different universities and countries within Africa could provide deeper insights into regional differences and help tailor interventions more effectively.

## CONCLUSION

This study provides valuable insights into the quality of life among African medical and health science students, emphasizing key demographic and academic factors influencing their well-being. The findings indicate that dentistry students generally experience higher QoL across multiple domains, which contrasts with students from other disciplines, such as pharmacy and nursing. Age and academic year were also significant predictors of QoL, with younger students and those in their initial years of study reporting lower scores. These results align with existing literature and underscore the need for targeted interventions to support students, particularly those in high-stress disciplines or at earlier stages of their education.

## ABBREVIATIONS

**IUA**      International University of Africa
**QoL**      Quality of Life
**WHOQOL-BREF**   World Health Organization Quality of Life-26 item tool

### Funding

The work was supported by the Deanship of Scientific Research at Majmaah University, Al Majmaah, KSA, under the Project No. R-2024-272. The funders had no role in study design, data collection and analysis, decision to publish, or preparation of the manuscript.

## Grant Disclosures

The following grant information was disclosed by the authors:
Deanship of Scientific Research: R-2024-272.

## Competing Interests

Faizan Zaffar Kashoo is an Academic Editor for PeerJ.

## Author Contributions

- Fatima Alzahra Galgam conceived and designed the experiments, performed the experiments, prepared figures and/or tables, authored or reviewed drafts of the article, and approved the final draft.
- Adil Abdalla conceived and designed the experiments, performed the experiments, prepared figures and/or tables, authored or reviewed drafts of the article, and approved the final draft.
- Mahmoud Shahin conceived and designed the experiments, prepared figures and/or tables, authored or reviewed drafts of the article, and approved the final draft.
- Magda Yousif conceived and designed the experiments, performed the experiments, prepared figures and/or tables, authored or reviewed drafts of the article, and approved the final draft.
- Nahla Abdulrahman conceived and designed the experiments, prepared figures and/or tables, authored or reviewed drafts of the article, and approved the final draft.
- Fatmah Alamoudi conceived and designed the experiments, performed the experiments, prepared figures and/or tables, authored or reviewed drafts of the article, and approved the final draft.
- Mehrunnisha Ahmad conceived and designed the experiments, prepared figures and/or tables, authored or reviewed drafts of the article, and approved the final draft.
- Amira Yahia performed the experiments, analyzed the data, prepared figures and/or tables, authored or reviewed drafts of the article, and approved the final draft.
- Mohammad Sidiq analyzed the data, prepared figures and/or tables, authored or reviewed drafts of the article, and approved the final draft.
- Aksh Chahal analyzed the data, prepared figures and/or tables, authored or reviewed drafts of the article, and approved the final draft.
- Fuzail Ahmad analyzed the data, prepared figures and/or tables, authored or reviewed drafts of the article, and approved the final draft.
- Mohammad Abu Shaphe analyzed the data, prepared figures and/or tables, authored or reviewed drafts of the article, and approved the final draft.
- Gopal Nambi analyzed the data, prepared figures and/or tables, authored or reviewed drafts of the article, and approved the final draft.
- Moattar Raza Rizvi analyzed the data, prepared figures and/or tables, authored or reviewed drafts of the article, and approved the final draft.
- Faizan Kashoo analyzed the data, prepared figures and/or tables, authored or reviewed drafts of the article, and approved the final draft.

## Ethics

The following information was supplied relating to ethical approvals (*i.e.*, approving body and any reference numbers):

The International University of Africa (IUA). Ethical approval was obtained from the Research Ethical Committee (REC) of the Faculty of Medicine at IUA (IRB-21).

## Data Availability

The raw data is available in the Supplemental File.

## Supplemental Information

Supplemental information for this article can be found online at http://dx.doi.org/10.7717/peerj.18809#supplemental-information.

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
