# Peer review of "Assessing the quality of life among African medical and health science students using the WHOQOL-BREF tool"

_PeerJ, doi:10.7717/peerj.18809_

## Round 0.1 · original submission · Minor Revisions

Three reviewers have evaluated your work and their feedback is generally positive. Minor revisions are required to further improve your manuscript. Please address all reviewers' comments and provide a point-by-point response with your revised submission.

·

Basic reporting

Abstract:
Methods: add details regarding the type of statistical tests done, and regarding significance of p value used.
Results: line 15 ‘revealed that dentistry students significantly predicted overall QoL (³ = 7.059, p < 0.05), as well’ what did it revealed statistically significant difference? Please add the suitable word.
Introduction:
Overall introduction is too short. Literature review needs to be added from the previously conducted using this study tool.
Why was this questionnaire specifically chosen? Are there other tools also available? Please add more details in introduction.
Methods;
What was the inclusion and exclusion criteria of the study?
Discussion:
Authors can remove the heading from discussion
Findings and comparison of domains of Qol should be added in comparison to the prior studies.

Experimental design

Rationale of the study needs to be better described. why is this study important?

Validity of the findings

no comments

Additional comments

no comments

Reviewer 2 ·

Basic reporting

The language needs revision to align with the standards of scientific writing. Ensure consistency, clarity, and grammatical correctness throughout the manuscript.
Some references (e.g., from 2000-2011) are outdated. Please update them with more recent literature to ensure the manuscript reflects the latest developments in the field.

Experimental design

the author need to write a research question

Validity of the findings

All data were provided

·

Basic reporting

Based on the initial write up of the manuscript, the authors were able to express their ideas clearly and used a standard way of reporting and language (English). Nevertheless, there were some few issues that were observed that need to be addressed in order for the manuscript to be much more reliable and credible.
1) Some of the mentioned literature references were quite obsolete. It would be much helpful if the authors will use a much more recent and relevant references, say 2015 and above will suffice. Everyday there are thousands of newly published being indexed in different databases like Google Scholar, Scopus, Web of Science, DOAJ, etc. So, updating some of the old literatures will not be a problem.
2) There were some missing tables for the presentation. The authors only mentioned or included tables and figures for the Mean Distribution of the QoL and Basic Demographic Characteristics (Frequency and Percentage). How about the result of the test of differences and the multiple regression analysis?

Experimental design

In order for a study to have a guide and flow, the authors should explicitly discuss or mention the research question of the study. Alternatively, they can also revise it to a Research Objective. In the study, the authors only shared their hypotheses, but how about mentioning other small relevant ideas like the level of QoL, and their Variances when grouped according to the demographic profile?
As for the Methods, there were some minor detail revision to be done by the authors:
1) While using a Cross-sectional study is common among research papers, it would better if the authors will provide relevant literature to support the design of the study.
2) The data collection period should be integrated in the Data Collection Technique for a more concise discussion.
3) The sample size and sampling technique can also be integrated into one subsection of the study (authors may simply choose one subheader between the two.).
4) While the study uses a purposive sampling method, it could benefit from a clearer explanation of why this method was chosen and how it affects the generalizability of the results. Additionally, discuss any potential limitations regarding the representativeness of the sample.

Validity of the findings

As mentioned from the previous comments, there were missing tables in the manuscripts that need to be included in the study. Specially the Multiple Regression wherein the authors hypothesized different elements that play a role in the QoL of the participants.
Since there were no clear research questions in the first part of the manuscript, the authors need to realign their conclusions based on the revised version.

Additional comments

Here are some additional comments and suggestions that the authors may consider:
1) Title: Please remove the "A Cross-Sectional Study" part, it is already implied in the study.
2) Abstract: Please include the sampling technique in the discussion as well as the statistical treatment.
3) Introduction: It is suggested to discuss and elaborate more regarding the subject matter of the study (QoL) by discussing comprehensively the major trends and issues about it from the global or regional context down to the national or local situationer. Strengthen your argument by citing relevant and recent literatures. Highlight the research gap and explain the rationale why is there a need to study such topic. Include the research questions or research objectives in the discussion as well. Lastly, provide discussion as to the contribution of this research paper to the body of knowledge and into the field of practice.
4) Results: Why is there a need to present the Reliability of the Instrument in the results wherein it should have been presented in the Methods section under the Data Collection tool? Remove this part and focus on other pertinent discussion of the study. Lastly, provide the missing tables for the test of differences and multiple regression analysis.
5) Discussion: Remove the discussion for the reliability of the instrument, it has no bearing for the study since it is a tool meant to be used repeatedly. Support your discussion with recent and relevant literature.

That is all. Thank you for the opportunity to review your manuscript. Good luck!

---

## Round 0.2 · accepted · Accept

All reviewers agree that you have adequately addressed their previous concerns and the manuscript has been substantially improved. The editorial office will contact you shortly regarding the next steps in the publication process.

·

Basic reporting

Authors have succesfully made all the changes I had requested.

Experimental design

Changes in the methods as suggested has been done

Validity of the findings

Changes made in the discussion section of the manuscript.

Additional comments

none

Reviewer 2 ·

Basic reporting

all the required amendments and recommendations have been appropriately addressed by the authors.

Experimental design

Not applicable

Validity of the findings

all underling data have been provided

·

Basic reporting

Most of the suggestions and recommendations were followed and the changes in the manuscript was substantial.
There is already enough literature references to support the conduct of the study as well as the argument and research problems were clearly established in the discussion. The methodology was also clear and can be replicated by someone if they wish to. The results and discussions are well and organized already. The message of the manuscript is presented appropriately.

Experimental design

The research is within the aims and scope of the journal. The research question was already explicitly mentioned in the early section of the manuscript (Introduction). Ethical standards and processes were followed successfully by the proponents.

Validity of the findings

The results of this study is significant and contributory to the field of practice and knowledge. The statistical treatments were appropriate and accurate. The conclusion were restructured and aligned with the study's research question.

Additional comments

Minor proofreading is advised but overall, I believe the manuscript is good and admissible by the journal.